# Strategies for robust renovation of residential buildings in Switzerland

Alina Galimshina [1] ✉, Maliki Moustapha[2], Alexander Hollberg [3], Sébastien Lasvaux[4], Bruno Sudret [2] & Guillaume Habert[1]

Building renovation is urgently required to reduce the environmental impact associated with the building stock. Typically, building renovation is performed by envelope insulation and/or changing the fossil-based heating system. The goal of this paper is to provide strategies for robust renovation considering uncertainties on the future evolution of climate, energy grid, and user behaviors, amongst others by applying life cycle assessment and life cycle cost analysis. The study includes identifying optimal renovation options for the envelope and heating systems for building representatives from all construction periods that are currently in need of renovation in Switzerland. The findings emphasize the paramount importance of heating system replacements across all construction periods. Notably, when incorporating bio-based insulation materials, a balance emerges between environmental impact reduction and low energy operation costs. This facilitates robust, equitable, and low-carbon transformations in Switzerland and similar Northern European contexts while avoiding a carbon spike due to the embodied carbon of the renovation.

Buildings and the construction sector represent more than 40% of greenhouse gas (GHG) emissions globally[1–3]. As the energy consumption of residential buildings in Switzerland is, on average, closely aligned with that of Northern European countries[4,5], it makes Switzerland an informative case study for exploring energy retrofitting scenarios in Europe. For many government and international organizations, the energy renovation of existing buildings is seen as a central strategy for the decarbonization of the building stock[6,7]. In Switzerland, with around 64% of buildings still heated by oil and gas and 70% of the building stock built before strong energy efficiency standards were implemented (Federal Statistical Office[8]), it is clear that fossil-based heating systems have to be removed[9].

Decarbonization of the building stock is usually only seen through decarbonization of energy systems combined with increased energy efficiency[10,11]. However, a growing number of studies point out that a high renovation rate can worsen the life-cycle climate impact if the embodied GHG emissions linked with the materials used for the renovation are not considered[3,12]. A fine line has then to be found between the necessary renovation of an energy-inefficient and fossil fuel-powered building stock and a deep renovation which will increase upfront GHG emissions[13] at the crucial moment where they need to drastically be reduced to avoid overshooting and overstepping planetary tipping points[14]. These additional upfront GHG emissions due to renovation are also referred to as 'carbon spike'[13].

To identify the optimal renovation solution considering the economic and environmental costs, optimization techniques with life cycle assessment (LCA) and life cycle cost analysis (LCCA) can be applied[15]. The advantage of LCA and LCCA is the consideration of the whole life cycle of a building from the material extraction to its end of life. However, when considering such a long period, many

[1]ETH Zürich, Institute of Construction and Infrastructure Management (IBI), Chair of Sustainable Construction, Stefano-Franscini-Platz 5, 8093 Zurich, Switzerland. [2]ETH Zürich, Institute of Structural Engineering (IBK), Chair of Risk, Safety and Uncertainty Quantification, Stefano-Franscini-Platz 5, 8093 Zurich, Switzerland. [3]Chalmers University of Technology, Department of Architecture and Civil Engineering, Sven Hultins Gata 6, 412 96 Gothenburg, Sweden. [4]University of Applied Sciences of Western Switzerland (HES-SO), School of Business and Management Vaud (HEIG-VD), Institute of Energies (IE), Avenue des Sports 20, Yverdon-les-Bains 1401, Switzerland. ✉e-mail: galimshina@ibi.baug.ethz.ch

uncertainties occur, which significantly affect the results of the LCA and LCCA[16–18]. Such uncertainties include the future climate, the effective service life of the materials, the future costs of energy, the user behavior, and the maintenance strategy of the owner, among others[19]. The GHG emissions associated with the materials' production also have high deviation due to variations in production facilities[20] as well as methodological assumptions during LCA calculation[21]. Such a combination of uncertainties significantly affects the resulting environmental impact and economic costs[22].

Several studies have previously considered building renovation strategies for one or several buildings to lower the overall GHG emissions and costs[23–25]. Others considered renovation strategies on a multi-building or building stock scale[26,27], and multi-objective optimization was applied in several studies to optimize the overall costs, emissions, or thermal comfort of the users[28–32]. However, the assessment of uncertainties is usually excluded. Only a few studies have included single or several parameters of uncertainty such as future electricity mix, climate change, replacement time, or material properties[33–35]. Our previous work has been focused on the combination of all the uncertainty sources during the building life cycle and robust multi-objective optimization for costs and environmental assessment[36], but the work was performed for only one case study of a residential building.

Conventional renovation solutions usually include fossil-based insulation such as polystyrene or aerogel. Such materials have high embodied GHG emissions due to their carbon-intensive production process and are not able to regulate humidity leading to low summer heat comfort[37]. Alternatives such as straw or hemp have been developed for decades[38–41] but are recently gaining scientific and industrial credibility[42–44], probably linked with the climate collapse approaching. Indeed, it is possible to renovate buildings with either external or internal insulation made of straw or hemp[45,46]. For instance, straw can be installed as strawbale or blown into wooden boxes later installed on the façade, and hemp can be used as hemp blocks or sprayed directly on site as hemp lime. Recent industrial development led to increased application. In this study, we test such non-conventional renovation techniques, and our results highlight the significant difference between conventional and non-conventional regenerative solutions.

In this study, we identify key strategies for the robust renovation of six Swiss buildings representing 70% of the existing Swiss residential building stock currently in need of renovation. The buildings are from the eRen project[47] where 193 buildings have been analyzed to define representative buildings. Using these buildings, we test the previously identified renovation strategies in terms of cost and environmental impact considering all possible future uncertainties affecting the results. Surrogate modeling with *polynomial chaos expansion (PCE)*[48,49] is applied for the uncertainty quantification to ease the computational burden of the analysis in comparison with Monte Carlo simulation. Uncertainty quantification (UQ) using Monte Carlo simulation has a slow convergence rate in cases where many parameters are considered in the model[50,51]. The scientific contribution of the paper is the inclusion of the sources of uncertainty associated with the building life cycle and the identification of the optimal and robust building renovation scenario for the building representatives of all the construction periods in Switzerland that currently require renovation. In addition, non-conventional renovation techniques with bio-based materials usually discarded from analysis are included.

The results were achieved using multi-objective optimization, considering future uncertainties to provide the most robust solution. The results confirm that the heating system is the most influential and robust parameter and should be prioritized in building renovation. However, our study unveils a significant contrast in the optimal solutions between using conventional and non-conventional insulation materials. Both involve the replacement of fossil heating systems for renewable energy sources for heating. However, when using conventional insulation material, the optimization leads to minimum insulation thicknesses, while non-conventional materials lead to thicker optimum insulation. This connects to the questions of social justice related to the choice of material in renovation. Sticking with conventional fossil-based insulation materials leads to a high energy bill for inhabitants (but using renewable energy systems) if the minimum insulation thickness is used. If so-called deep renovation with thick conventional insulation is applied (as most current energy policies would lead to), the energy costs are reduced, but high upfront embodied GHG emissions are leading to a 'carbon spike'. On the contrary, using non-conventional bio-based materials such as straw or hemp leads both to a low GHG emissions impact thanks to carbon stored in the material and to a low energy bill thanks to high insulation thickness.

## Results

The results of the optimal solutions for the different heating systems considering conventional and non-conventional materials can be seen in Figs. 1 and 2. Regarding the LCA results, all the solutions lower the GHG emissions of the non-renovated building. In general, as expected, the solutions with the highest GHG emissions include the gas boiler and conventional materials while the most climate-friendly solutions include the wood pellets boiler or heat pump with a high amount of non-conventional insulation. A clear difference between the heating demand of conventional materials and non-conventional materials can be observed. The optimized solutions with non-conventional materials show higher insulation thicknesses and, therefore, lower energy consumption for heating and lower GWP, which indicates that such solutions are not only climate-friendly but also reduce energy consumption. It can also be noticed that solutions with wood pellet boilers and heat pumps show higher robustness in comparison to gas boiler solutions. The resulting lower GWP of the solutions with non-conventional materials can also be explained by the low embodied GWP of non-conventional materials and thermal properties provided by the high level of insulation. It is still clearly noticeable that the replacement of the fossil heating system has the highest potential to decrease the impact. In most cases, the optimal solution does not prescribe the change of windows, which can be explained by the high embodied carbon and investment costs associated with the installation of the new windows as well as the initial thermal performance of the existing windows. The resulting renovation solutions for median solutions are presented in Supplementary Information (SI), Section 6.

Figure 2 presents the results of the heating demand and LCCA. No clear trends can be observed as the solutions are generally in the same range of costs. The renovation of the envelope does not change the overall life cycle costs but shifts the share of the operational and investment costs.

It can also be noticed that the results for the LCCA are considerably less robust than the LCA results, which can be explained by the higher fluctuation of costs during the building life cycle. In general, similar trends for all the construction periods can be noted. The renovation materials with the resulting average life cycle costs and GWP for each of the presented solutions can be found in the SI (Section 6).

The results were also summarized in comparison of the initial heating demand to GWP and life cycle costs after renovation considering conventional and non-conventional materials (see Fig. 3). The results for the wood boiler are shown, while a summary for other systems can be seen in the SI (Section 7).

As can be seen in Fig. 3, a clear differentiation between conventional and non-conventional materials can be observed. The values for GWP for non-conventional materials are three times lower on average. It can also be noted that the uncertainty ranges do not overlap, which expresses no probability of achieving the same overall GHG emissions and shows that the non-conventional solutions perform better under all circumstances.

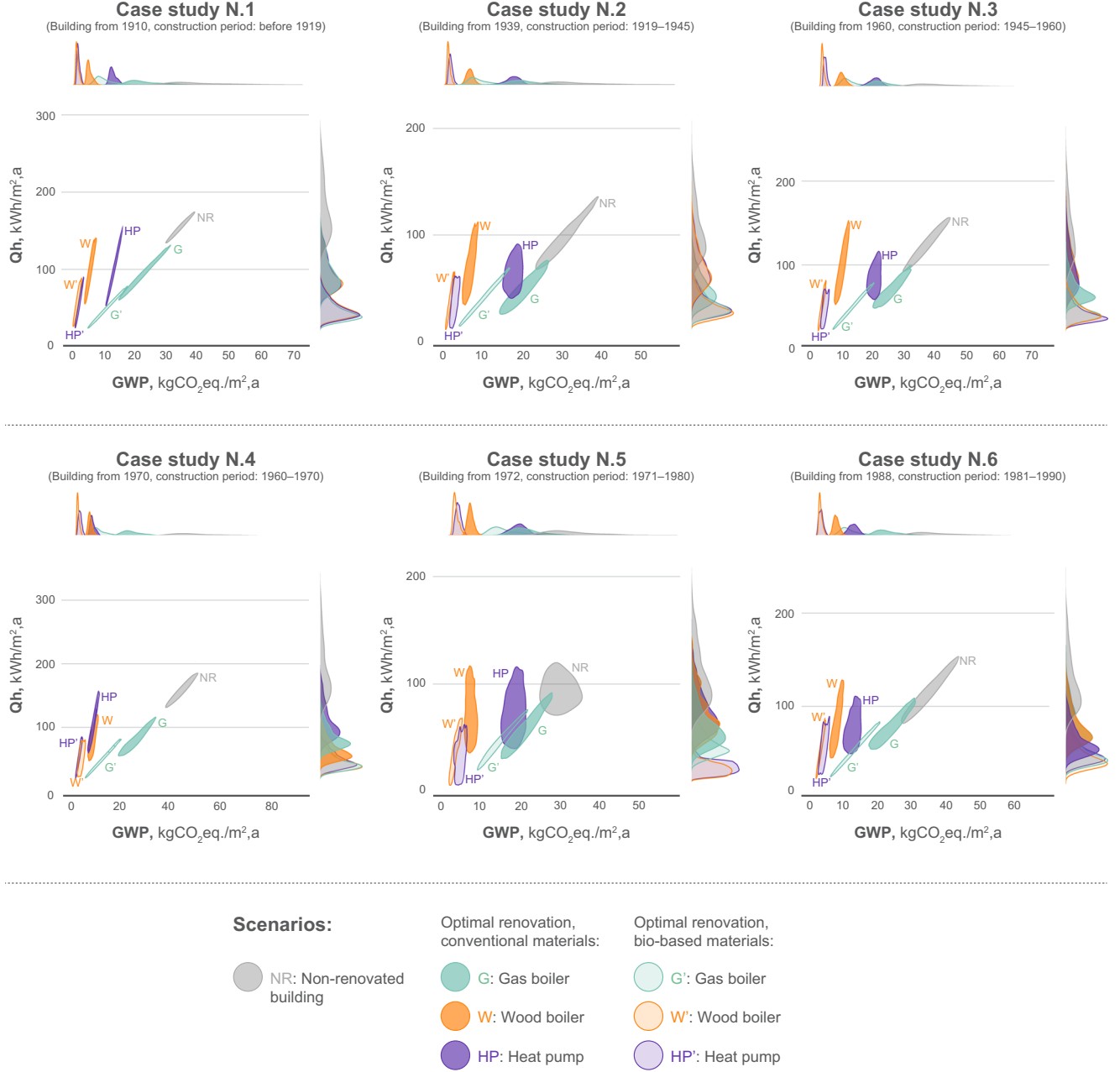

**Fig. 1 | Comparison of the heating demand (Qh) with global warming potential (GWP) of the optimal solutions.** Six case studies from different construction periods are presented, and solutions with three different heating sources are explored. Conventional and non-conventional materials are considered and compared to the non-renovated building. The results were obtained through the multi-objective robust optimization using a non-dominated sorting genetic algorithm (NSGA). The distributions of the solutions represent the 5th and 95th percentiles. Source data are provided as a Source Data file.

With regards to the LCCA, the results of non-conventional materials show lower life cycle costs on average for all the buildings. However, considering the uncertainties, the solutions represent a high probability of overlapping, which does not ensure the benefit of using non-conventional materials in terms of costs. However, a clear difference can be seen once the initial heating demand is higher than 150 kWh/m² a. In these cases, the operational savings pay off the initial investments.

The results of life cycle costs and GWP for the identified optimum solutions are shown separately for the production and operational stages (Figs. 4 and 5). The investment costs for non-conventional renovation are clearly higher than for conventional renovation. However, considerable savings on the operational costs can be observed. It can also be noticed that the share of the increased investment costs for

the non-conventional solution compared to the conventional materials is, in most cases, lower than the share of the operational savings. With regards to LCA, it can be clearly seen that the conventional deep renovation without fossil-based heating system replacement is not beneficial in terms of life cycle GHG emissions.

Considering the optimal amount of insulation material for the external walls, a clear separation is visible (Table 1). In most of the cases, the maximum thickness of non-conventional materials is identified while the range of optimal results for conventional materials lies within 0–10 cm. Looking at the whole building envelope, in very rare cases, the insulation thickness exceeds 20 cm.

In the majority of the non-conventional cases, the optimal solution includes 70 cm of straw bale insulation on the exterior wall, which is the maximum our model is allowed to consider. The same amount of

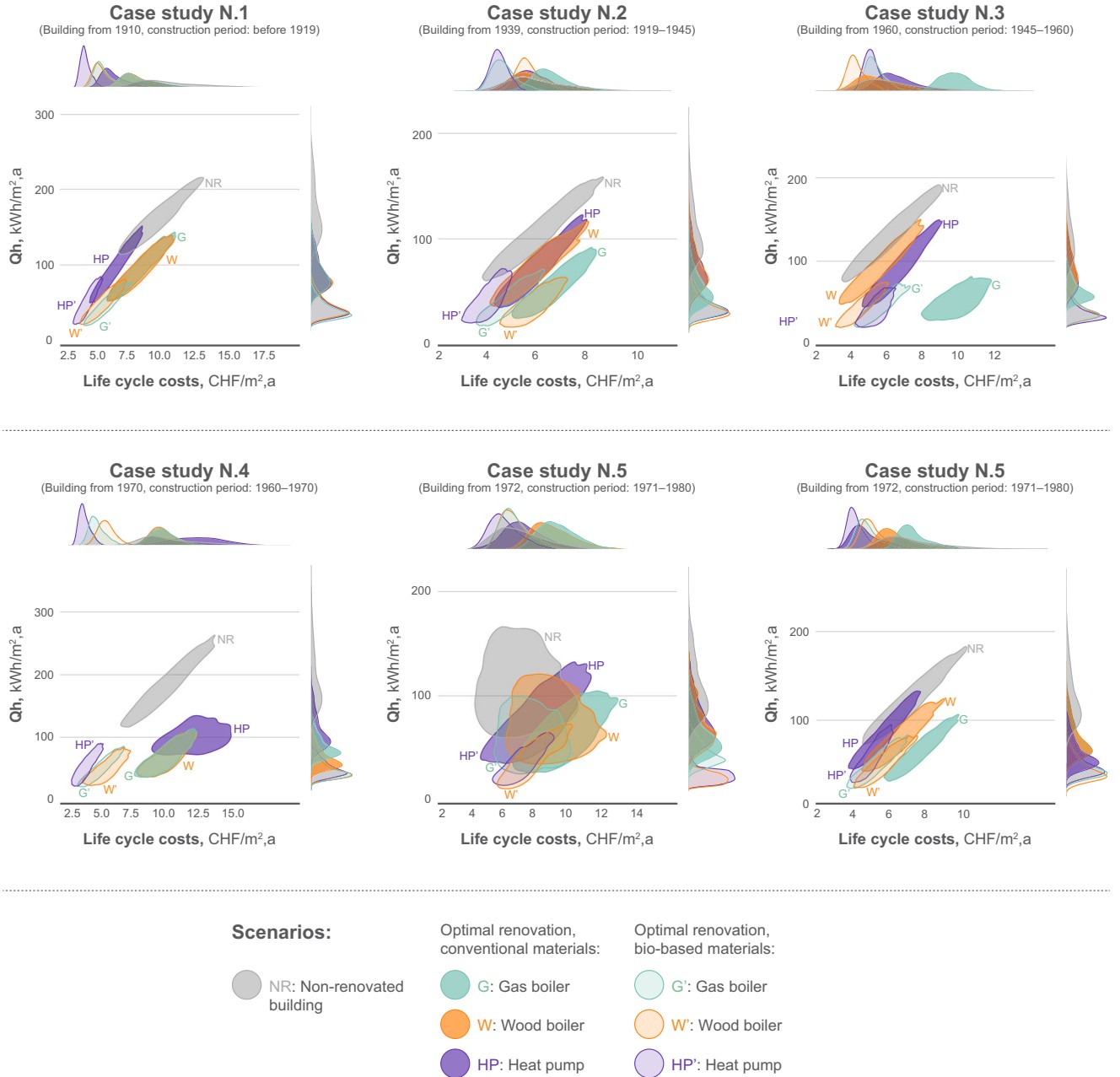

**Fig. 2 | Comparison of the heating demand (Qh) with life cycle costs of the optimal solutions.** Similarly to the Fig. 1, six case studies from different construction periods are presented and solutions with three different heating sources are explored. Both conventional and non-conventional materials are examined and compared to the non-renovated building as a reference. The outcomes were derived through multi-objective robust optimization employing the non-dominated sorting genetic algorithm (NSGA). The distributions of the solutions represent the 5th and 95th percentiles. Source data are provided as a Source Data file.

insulation is obtained for the roof. Considering the moisture safety and risk of mold growth, only EPS and hempcrete were applied on the ground floor. Therefore, the solutions for the ground floor differ from the exterior wall and roof.

The replacement of windows occurs only in three out of 18 scenarios for conventional and in one out of 18 scenarios for non-conventional materials.

## Discussion
In this work, optimal building renovation solutions were identified for the building representatives from the construction periods that currently require renovation. To identify how representative the case studies are for the whole building stock, an additional study was

performed where the heating demand of the selected building representatives was scaled up according to the energy reference area associated with each construction period[4]. This allowed estimating the heating demand (kWh/a) for each construction period. The results were compared with the actual energy consumption for heating buildings constructed before 2000[52] The analysis showed that the resulting values differ by 4%. Therefore, the results of the study can be generalized for the residential building stock in Switzerland. Considering that, an additional follow-up study was performed to evaluate the amount of saved GHG emissions in case the optimal renovation scenarios were applied for the whole residential building stock in Switzerland. The results show that the solutions with either heat pumps or wood pellets with the inclusion of non-conventional

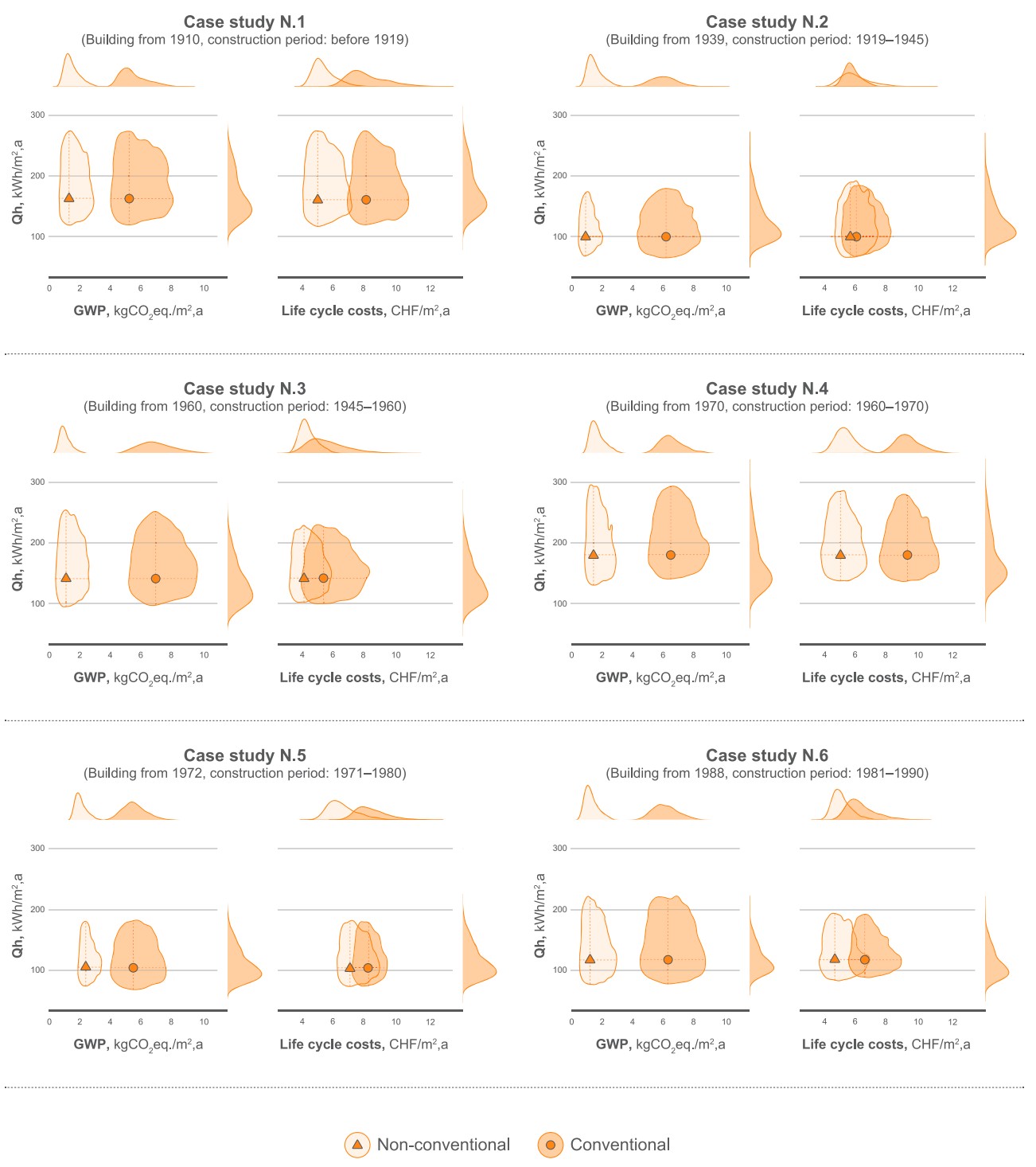

**Fig. 3 | Initial heating demand (Qh) and results for GWP and life cycle costs after renovation.** The orange dot and triangle represent the mean value for the Qh and GWP or Qh and Life cycle costs for the conventional and non-conventional materials, respectively. The distribution represents the 5th and 95th percentiles. Source data are provided as a Source Data file.

materials allow saving up to 87% of the GHG emissions per year. The analyses for both generalizations of the building stock and potential savings from the optimal renovation scenarios can be found in SI (Section 5).

This study confirms that the most influential parameter in building renovation is the heating system which allows a significant reduction of GHG emissions, which had been stated previously[19]. However, besides the reduction of GHG emissions, it is important to consider the potential energy reduction and social justice questions. When

maximizing the amount of fast-growing non-conventional insulation materials, it is possible to store carbon and reduce the energy bill of the residents of the building.

It is important to note that this study primarily aimed to identify optimal solutions based on GHG emissions and costs. While legal thermal retrofit requirements exist in Switzerland, the objective was to explore whether energy renovation aligns with climate goals. The results reveal significant differences between the two approaches. While conventional deep renovation without fossil heating system

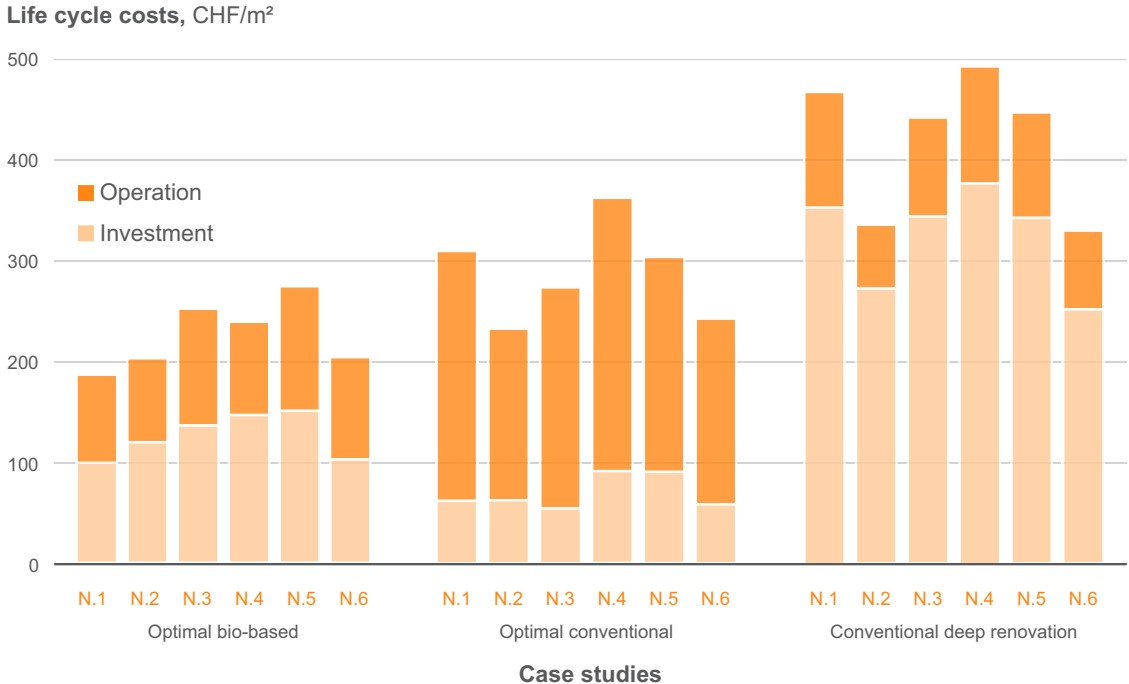

**Life cycle costs**, CHF/m²

**Fig. 4 | Investment and operational costs for the examined case studies considering conventional and non-conventional materials with wood boiler as well as conventional deep renovation with gas boiler.** The average values are shown, and the replacement costs are excluded. Source data are provided as a Source Data file.

replacement complies with the energy retrofit regulations, it increases the embodied impact significantly and does not yield an overall reduction in GHG emissions across the building's lifecycle. Consequently, this underscores that the current building standards should be updated towards the reduction of GHG emissions, which is the priority to reach carbon neutrality by 2050. The only approach that satisfies both energy retrofitting mandates and leads to a reduction in lifecycle GHG emissions is the integration of thick bio-based insulation coupled with the replacement of fossil-based heating systems.

This study highlights a significant contrast in the optimal insulation thickness between conventional and non-conventional materials. The rationale behind the optimal insulation thickness can also be attributed to the resulting carbon intensity of the electricity grid. Switzerland's consumer mix, for the most part, has a low carbon footprint (0.125 kgCO₂eq./kWh). This, in turn, leads to a preference for thinner insulation when using carbon-intensive materials in combination with a heat pump. However, bio-based materials possess much less embodied carbon, and the consequent carbon payback time is much lower compared to the payback of conventional materials. The results of this work are applicable to countries with similar grid intensity. The EU strategy on the decarbonization of the electricity grid in line with the European Union Green Deal objectives would lead to grid carbon intensities similar to the Swiss ones. Therefore, the results of this study can also be projected for heating-dominated European countries[53] (European Commission[7]; European Environment Agency[54]). It is crucial to note that the decarbonization of the electricity grid will elevate the significance of embodied GHG emissions and bring this stage to the most influential in the building life cycle. Therefore, the actions on adapting the low carbon non-conventional materials must be implemented.

Renovation scenarios with bio-based materials necessarily involve a question of resource availability for a potential upscale. Contrary to most studies focusing on structural timber construction[55] and showing the limit of current availability[56] or the necessary land use changes to operate to grow timber cities[57], renovation with bio-based material has much less land use consequences. It has been shown that the amount of straw needed for the renovation of the whole EU building stock would consume a maximum of 10% of the wheat straw that can be sustainably removed from the land[58]. Using straw as insulation material instead of burning it and thereby sequestering carbon within the building envelope can offset up to 3% of the overall GHG emissions from all sectors[59].

Concerning the availability of wood as an energy source for wood boilers or district heating systems, a recent report provides estimates for wood energy potential in Switzerland[60]. The range varies from a sustainable level of 2,500 GWh/a considering ecological, economic, legal, and political constraints to a theoretical level of 25,100 GWh/a considering the total amount of wood available as an energy source.

Scaling up the optimal solution that utilizes wood for heating alongside thick bio-based insulation to the entire building stock would require an estimated 24,000 GWh/year. However, only 11% of the total demand could be covered, considering economic, ecological, and social restrictions. This implies that a combination of wood-based heating (as small-scale individual boilers or large-scale boilers in district heating) and heat pumps would be necessary to upscale the solution proposed by this study to the entire building stock.

A second potential challenge of using multiple individual wood boilers in densely populated areas is particle emissions that can significantly impact air quality and human health. The quantity of particulate emissions highly depends on the quality of the combustion process. The amount of pollutants in the modern-type automatic boiler is much lower and supported by European countries[61]. In addition, the results for the individual boilers in this study can be transferred to district heating which can filter out the particles because the GHG intensity is comparable.

## Limitations

In this work, building renovation was limited to envelope insulation, windows, and heating system replacement. To fully assess the possible renovation scenarios, renewable energy measures such as solar photovoltaics (PV) could be added in the future. It has been shown in previous studies that once the PV panels are added along with the other renovation options, combined window-façade insulation might appear as an optimal renovation option[62]. The study explored the

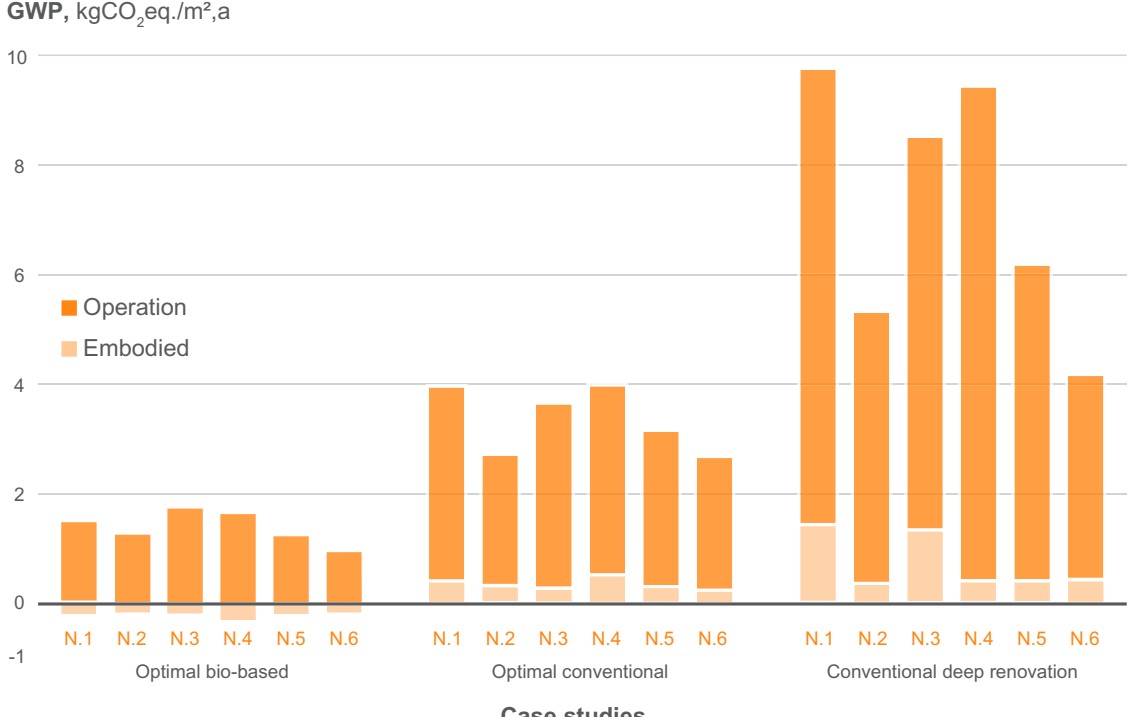

**Fig. 5 | Embodied and operational GHG emissions for the examined case studies considering conventional and non-conventional material with wood boiler as well as conventional deep renovation with existing gas boiler.** The deterministic values are shown, and the emissions related to replacement are excluded. Source data are provided as a Source Data file.

**Table 1 | Optimal thickness of thermal insulation for external walls after renovation considering non-conventional and conventional materials**

| Building type | 1911 (building 1) | 1939 (building 2) | 1960 (building 3) | 1970 (building 4) | 1972 (building 5) | 1988 (building 6) |
|---|---|---|---|---|---|---|
| Conventional materials | 10 cm | 0 cm | 0 cm | 10 cm | 5 cm | 7 cm |
| Non-conventional materials | 70 cm | 70 cm | 70 cm | 70 cm | 20 cm | 70 cm |

potential of using thick, fast-growing bio-based insulation. Such thick insulated walls cannot be implemented everywhere, especially in urban contexts where space might be limited. However, many recent projects highlight the feasibility of thick straw bale walls, such as a 50 cm wall in Paris city center and architectural projects with 0.8–1.2 m walls in Switzerland[63,64].

Only individual residential heat pumps were considered in this work. However, practical constraints such as limited space or esthetic considerations could sometimes prevent their applicability, and the use of the heat pumps applied by a district heating network would be required. Due to the energy crisis in 2022, energy prices and discount rates have changed recently and could potentially result in alterations to the outcomes. Therefore, they should be updated in the model for future studies.

Another limitation of the study is that the current assessment is based on LCI data encompassing both production and end-of-life stages of LCA. Nevertheless, the outcomes do not differentiate the distinct impacts of these stages when considered individually, potentially influencing the study's findings.

We used a quasi-steady analysis of the energy performance to account for the operational GHG assessment and costs, which is the standard in Switzerland and many European countries. It has also been shown that monthly analysis is performing equally well as the hourly model for heating[65]. However, to increase the accuracy of the model,

dynamic energy analysis could be used to simulate overheating hours and potential cooling needs.

## Recommendations

In this work, the methodology for robust and optimal building renovation was applied to six residential building representatives covering all construction periods in Switzerland that are currently in need of renovation. Clear recommendations for the future implementation of building renovation for Swiss residential buildings can be drawn, which can be expanded to other industrialized countries where renovation is the main focus:

1.  Fossil heating system replacement is the most influential parameter in building renovation to decrease the GHG emissions attributed to the building sector.
2.  When non-conventional, fast-growing bio-based materials are considered, the maximum possible amount of insulation material should be applied in combination with replacing the fossil heating system. Such solutions do not only drastically reduce the GHG emissions of a building, but also store carbon and reduce the energy bill of the residents due to the reduced operational energy consumption.
3.  When conventional materials are considered, an optimal, robust, cost-effective, and climate-friendly solution does not prescribe deep renovation. Only a small amount of insulation on the

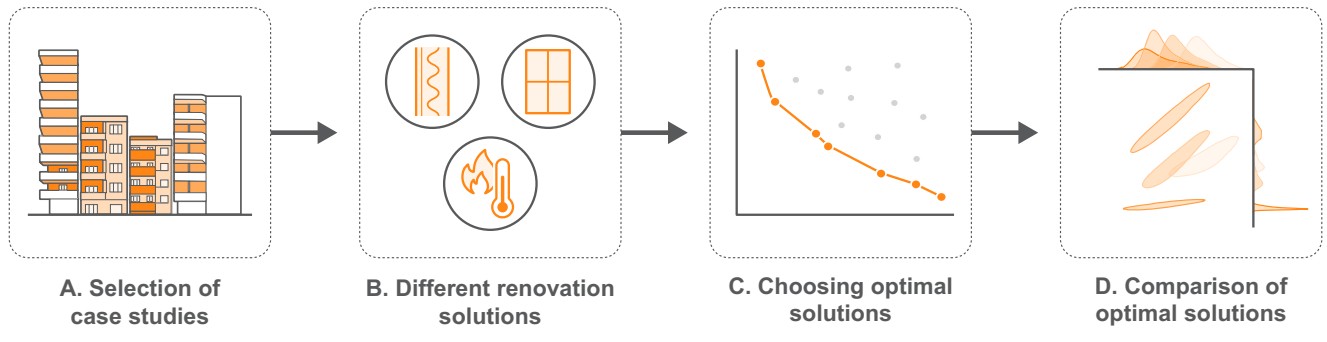

**Fig. 6 | Methodology of the paper.** The illustrations of the buildings are adapted from the eRen project[47].

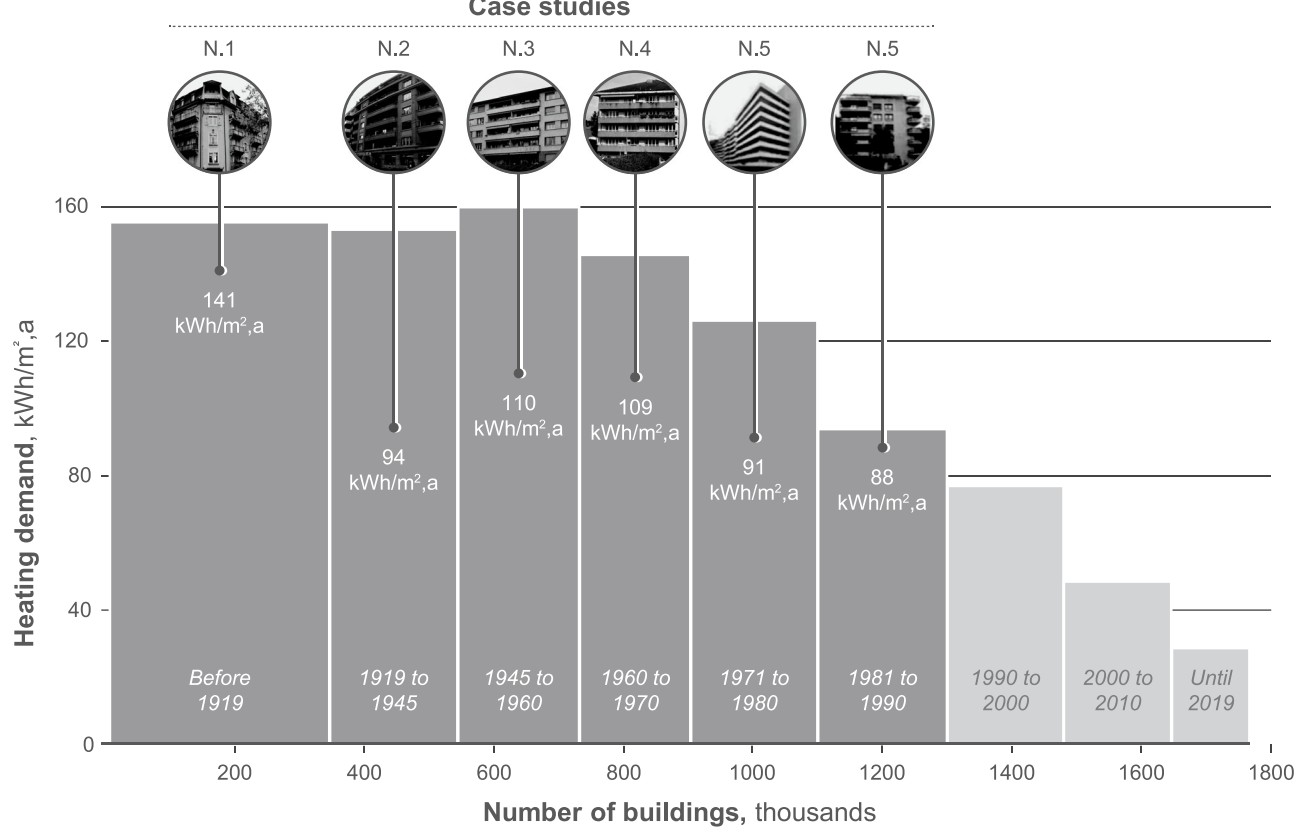

**Fig. 7 | Swiss building stock according to their year of construction (Federal Statistical Office[77]) with the case studies analyzed in this paper and their heating demand (N.1–N.6).** Source data are provided as a Source Data file. The photos of the buildings were obtained from the eRen project[47].

exterior wall is beneficial from a life cycle perspective. However, this leads to a higher operational energy consumption.

4. Only in the case that the initial heating demand of a non-renovated building is higher than 150 kWh/m²a on average renovation pays off economically. This only happens when the renovation is carried out with non-conventional materials and without window replacement. This can be explained by the low energy price for fossil fuels, which in turn explains the problem of the low renovation rate. However, the growing costs for fossil fuels might change the perspective on building renovation and fossil heating systems' replacement.

## Methods
Figure 6 describes the methodology of the paper. First, residential building representatives were identified for each construction period where building renovation is currently required. Second, renovation options considering conventional and non-conventional materials

were selected. Afterward, an integrated assessment of costs and environmental impacts was created, and robust optimization was performed to identify the most cost-effective, climate-friendly, and robust solutions for building renovation. These solutions are then compared in a probabilistic context. In the following, each step is explained in detail.

### Selection of case studies
To identify the building representatives, the models from the eRen project were used, where 15 building models were defined to represent the residential building stock for multi-family houses in Western Switzerland[47]. The project is based on data from 193 buildings in Fribourg, Vaud, and Geneva cantons, collectively encapsulating between 72% and 89% of Switzerland's dwelling count, according to the Federal Statistical Office (FSO) (Federal Statistical Office[66]). Six buildings representing the construction periods currently in need of renovation were selected for this study, which is shown in Fig. 7. The presented

case studies have different energy reference areas (ERA) as well as different structural materials, e.g., reinforced concrete, hollow bricks, or stone. For several building representatives, envelope insulation had been applied earlier, mostly to the roof structure, which was considered as an initial state in this study. The presented buildings have different energy performances and give an idea of the energy performance of buildings over the construction periods. Prior research extensively characterized the thermal performance and retrofit status of Swiss residential buildings using Swiss Cantonal Building Energy Certificates and highlighted the variability within the building stock[67]. The case studies presented in our work notably represent buildings with the median energy performance within the building stock, predominantly lacking insulation, and cover most of the operational GHG emissions of buildings in Switzerland while being supplied by oil and gas. The buildings' structure and energy performance can be seen in the SI (Section 1) of this paper.

### Renovation solutions

Heating system replacement, thermal envelope insulation, and window replacement were considered in this work. Renewable energy production, such as photovoltaics, was not included as it can be considered as an energy infrastructure transition question rather than a building renovation question[57]. In general, two options for insulation materials were applied−conventional materials and fast-growing, bio-based materials, which are referred to as non-conventional in this study. Conventional renovation options are represented by the materials that are commonly applied in practice, such as EPS, glass wool, rock wool, and cellulose fibers. Non-conventional materials are straw bales, hemp mats, and hempcrete. Concerning window replacement, double and triple-glazing options with aluminum, PVC, and wooden frames were considered. The materials and their properties are presented in the SI (Section 3).

In the case of non-conventional renovation solutions, biogenic carbon sequestration was accounted for. A dynamic carbon storage assessment was applied where time-dependent characterization factors were used, and KBOB values were applied for the carbon release[68].

### Integrated assessment of LCA and LCCA, uncertain parameters, and robust optimization

To analyze the overall GHG emissions and costs, an integrated analysis of LCA and LCCA was performed. The stages production (A1–A3), replacement (B4), operational energy demand (B6), and end-of-life (C3–C4) were used in LCA with reference to the scheme of EN 15978:2012. In LCCA, the investment, operational, and replacement costs were included. Repair was taken into account for each year as a percentage of the investment costs. The reference study period was set to 60 years according to the Swiss standards[69]. The functional unit for this study encompasses the complete life cycle of a renovation of a residential building operating for 60 years, including construction, occupancy, maintenance, and potential replacement of building components. It evaluates the environmental impacts associated with the building's production, replacement, operational energy demand for heating, and end-of-life stages, considering the use of both conventional and non-conventional materials. The only impact category considered is climate change. Global Warming Potential (GWP) expressed in $kgCO_2$eq. is used as the indicator. The embodied carbon values for stages A1–A3 and C3–C4 are taken from the KBOB 2016 database[70]. To account for the operational stage, the energy demand for heating was calculated using a monthly quasi-steady-state analysis according to the Swiss standard[71,72]. The resulting useful heating demand was multiplied by an efficiency factor to calculate the final energy demand, which was multiplied with GWP values from the KBOB database depending on the respective energy carrier used (gas, wood, electricity, etc.).

The uncertain parameters can be classified in different ways. In this work, the uncertainties were separated into those that can be affected by the designer (design parameters) and those that cannot be affected but need to be carefully considered (exogenous parameters). Design parameters represent the type of insulation or heating system in place. Exogenous parameters are intrinsic and scenario parameters in the model, such as climate change, the replacement time of the building materials, future electricity mix in Switzerland and associated operational costs and impacts, occupancy behavior, and inflation rate to account for the price fluctuation of the future energy and material costs. The parameters' description, including range and distribution are shown in the SI (Section 2). The assessment of the future climate can be found in Galimshina et al.[72].

We perform a multi-objective robust optimization where we identify the optimal renovation solution to minimize the two objectives of LCA and LCCA, considering the associated uncertainties. We use the non-dominated sorting genetic algorithm II (NSGA-II) which is in particular useful while dealing with the combination of discrete and continuous parameters. Discrete design parameters are possible renovation options, and continuous parameters can take any value within a defined range, for example, the insulation thickness. Several studies have used NSGA-II for building renovation[73,74]. The main drawback of NSGA-II is the associated computational cost, especially considering the uncertain parameters. To overcome this, we use surrogate modeling techniques. Several techniques for surrogate models can be used, for instance, Polynomial Chaos Expansion or Kriging. In this work, we use the methodology of coupling NSGA-II with Kriging as a Gaussian regression process for surrogate modeling as proposed by Moustapha et al.[75]. The detailed methodology can be seen in the SI (Section 4).

### Probabilistic comparison

Once the optimization results were obtained, the Pareto front of the optimal solutions considering life cycle costs and life cycle GHG emissions was plotted. The median solutions for each heating system type were compared separately in a probabilistic context. The solutions were also compared to the non-renovated building results and conventional renovation solutions. The 5th and 95th percentiles were used for the probability range.

## Data availability

The data supporting the findings of this paper can be found in the Supplementary Information, Source Data, and accompanying code provided with this paper. Source data are provided in this paper.

## Code availability

The code is uploaded within the Code Ocean platform[76].

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

## Acknowledgements

The research was funded by the Swiss National Science Foundation, project number 172545. The authors would also like to thank Diogo Guerra for his invaluable help refining the figures presented in this paper.

## Author contributions

Alina Galimshina: Conceptualization, Methodology, Software, Writing—Original Draft. Maliki Moustapha: Methodology, Software, Writing—Review & Editing. Alexander Hollberg: Conceptualization, Methodology, Supervision, Writing—Review & Editing. Sébastien Lasvaux: Methodology, Resources, Writing—Review & Editing. Bruno Sudret: Methodology, Software, Supervision, Writing - Review & Editing. Guillaume Habert: Conceptualization, Methodology, Supervision, Writing—Review & Editing.

## Competing interests

The authors declare no competing interests.
