## [Peer Review File · Nature Communications]

Strategies for robust renovation of residential buildings in SwitzerlandREVIEWER COMMENTS

Reviewer #1 (Remarks to the Author):

The study provides recommendations for robust building renovation strategies considering uncertainties on future evolution of climate, energy grid, user behaviours. The research is interesting while many issues exist in the current version of paper.

1. Recommendations is not a specific enough keyword for using in title.
2. The quality of the figures in the paper need to be upgraded to meet top level journal standard.
3. Methodology description should be extended.
4. I did not see detailed evaluation of uncertainty in the paper content.
5. The scientific contribution and renovation of this research is not clear.
6. Whether the recommendations are fit for buildings in other regions and countries? The representativeness of the case study? All these questions need to be discussed.

Reviewer #2 (Remarks to the Author):

This paper addresses an interesting and relevant topic and it is generally well written. Nevertheless I struggle to understand essential aspects of the applied method and of the results. My questions/comments are:

- Is the absence of any regulation (legislation) for retrofitting assumed or are the authors establishing the best LCA and LCC results for compliance with existing regulation on thermal retrofit. This should be clarified in the text. In the latter case it would also need to be specified what the targeted thermal performance is (in kWh/m²/year). In the former case (absence of any regulation) it should be clarified which thermal performance is reached (final energy consumption by building type).
- It is not clear to me why cellulose fibers are considered as conventional materials in spite of being typically produced from recycled paper.
- Which assumptions were made for cladding/exterior wall cover (which technique/s, materials etc.)? Considering the content of the supplementary material, I get the impression that cladding/exterior wall cover was neglected which would be a very rough simplification both for LCA and LCC. If voluminous bio-based materials are used this requires also more material for the substructure. Was this taken into account? An insulation thickness of 70 cm is likely to be impossible due to space constraints (e.g. in urban context). Wherever it IS possible it is likely to call for significant changes to the building, e.g. extension of the roof as a consequence of the thicker walls, larger windows to ensure the same amount of natural light in the interior etc. Were these aspects taken into account, including both the significant LCA and LCC implications of? Moreover, for a significant share of the building stock, the exterior appearance must not be changed (heritage protection). Was this considered?
- In the case of conventional materials, the insulation thickness of the optimal solutions is amazingly low while the thickness of the bio-based insulation materials is surprisingly high (very striking also acc. to Fig.6 of the appendix). Is this the outcome of the LCC or the LCA calculations? Is the reason for this outcome explained anywhere? (I don't think so) Could it be that the cost of the bio-based solutions is significantly underestimated (next to ignoring other constraints) which would explain the very large insulation thicknesses and the high energy savings?
- Why is window replacement not recommended by the model while other models do so and while window replacement is also quite common in reality?
- Which energy prices were assumed? Do the operational cost for heating in the supplementary material refer to 1 kWh of final energy of 1 kWh of useful energy (heating service)? Do the energy prices for fossil fuels include the CO₂ levy?

In any case, the range of energy prices seems very/too narrow.

- Which end-of life waste management treatment technology was assumed for conventional and non-conventional materials? How does this life cycle stage contribute to the final results?
- Since existing buildings are mostly heated with oil and gas, the yellow areas (representing non-renovated buildings) would be expected to be much wider in Fig.4.
- Which emission intensity was assumed for the electricity used for operating heat pumps? In more detail, do the values in the supplementary material refer to 1kWh of final energy of 1 kWh of useful energy? Is only domestically produced electricity and wood assumed to be used? (in winter, a significant share of the electricity is imported).
- The data sheet in the supplementary material lacks clarity, e.g. is "Cost gas boiler" = investment cost?. If so, boiler sizing does not seem to depend on the insulation level which would be questionable. Similarly, mustn't the embodied impact of the boilers (in kg CO₂ eq.) depend on their size (and in turn depend on the insulation level)?
- The discussion section refers to a follow-up study on optimal renovation scenarios but isn't this the content of the present paper?
- Were always individual heat pumps assumed (this is not always possible due to space and noise constraints and for aesthetic reasons)?
- In many large urban agglomerations, heating with wood is restricted or forbidden. It seems this was not considered.
- Does the analysis focus on space heating or is hot water supply also included?
- How exactly is the functional unit defined? (the functional unit should describe in quantitative terms the primary function/s fulfilled)
- If the large-scale use of bio-based materials is seriously recommended both as insulation material and for heating, the availability of these types of materials should be compared to the potential demand.
- Given surging prices for energy, materials and equipment as well high inflation rates in Europe, at least some words of warning would be required. This also concerns the discount rate which would need to be higher for high inflation rates.

Response to referees

Article title - Strategies for robust renovation of residential buildings – A Swiss focus

Referee 1:

1. Recommendations is not a specific enough keyword for using in title. →

Thank you for the comment, indeed, the title might not be specific enough. Here we propose another title: “Strategies for robust renovation of residential buildings – A Swiss focus”. The title has been adapted in the corrected version of the paper.

2. The quality of the figures in the paper need to be upgraded to meet top level journal standard.

Thank you for the comment, we extensively worked on the figures quality and revised them.

3. Methodology description should be extended.

Thank you for the comment. The methodology description was shortened due to the constraints of the size of the paper. However, specific aspects such as the renovation scenarios and probabilistic assessment were provided in more detail within the supplementary information.

4. I did not see detailed evaluation of uncertainty in the paper content.

Thank you for the comment, the methodology of robust optimization including a detailed uncertainty evaluation was added to the supplementary information due to the space constraints in the main content of the paper.

5. The scientific contribution and renovation of this research is not clear.

Thank you for your comment. The scientific contribution of the paper is the inclusion of the sources of uncertainty associated with the building life cycle and the identification of the optimal and robust building renovation scenario for the building-representatives of all the construction periods in Switzerland that currently require renovation. By including the sources of uncertainty, the study acknowledges the variability in building renovation projects, providing a more realistic assessment of their outcomes than previous studies. The definition and description of uncertainty sources also allows to identify the most influential parameters in building renovation. The description has also been added to the paper.

6. Whether the recommendations are fit for buildings in other regions and countries? The representativeness of the case study? All these questions need to be discussed.

Thank you for the valuable comment.

The findings of this study hold relevance for countries characterized by comparable grid carbon intensity, specifically those currently engaged in mandatory thermal renovations, such as the Northern European countries. The rationale behind this can be attributed to the optimal insulation thickness identified in this work. While carbon intensity of the Swiss grid mix is considerably low (0.1 kgCO₂eq./kWh), the optimal approach for conventional solutions does not involve substantial insulation. Conversely, non-conventional, low-carbon solutions require thicker insulation due to their significantly reduced carbon payback time, contrasting the conventional materials. Given the alignment of the results with the European Union's strategy for electricity grid decarbonization, as outlined in the European Union Green Deal objectives, these findings can be projected to encompass the future trajectory of Northern European countries (European environment agency, 2023). It has also been shown that the

energy consumption of residential buildings in Switzerland is, on average, closely aligned with that of Northern European countries (Atanasiu et al., 2011; Streicher et al., 2019).

Referee 2

1. Is the absence of any regulation (legislation) for retrofitting assumed or are the authors establishing the best LCA and LCC results for compliance with existing regulation on thermal retrofit. This should be clarified in the text. In the latter case it would also need to specify what the targeted thermal performance is (in kWh/m²/year). In the former case (absence of any regulation) it should be clarified which thermal performance is reached (final energy consumption by building type).

Thank you for the comment. While thermal retrofit requirements exist in Switzerland, this study primarily aimed to identify optimal solutions based on carbon emissions and costs irrespective of current regulations. The objective was to explore whether energy renovation aligns with reaching carbon reduction goals. The results reveal significant differences between the two approaches. The results can also support adapting regulations in the future to be in line with carbon reduction goals.

The thermal performance of the optimal solutions was added to Supplementary information and can also be seen in Figures 1 and 2 and the discussion on the carbon and energy retrofit is added to the main content of the paper.

2. It is not clear to me why cellulose fibers are considered as conventional materials in spite of being typically produced from recycled paper.

Thank you for the comment, indeed, a clear explanation between these materials was not made. By conventional materials, we meant the ones that are commonly applied, therefore, wood fiber and cellulose fibers fall into that category. To enhance clarity, we have now relabeled bio-based materials as non-conventional materials. Your feedback is greatly appreciated.

3. Which assumptions were made for cladding/exterior wall cover (which technique/s, materials etc.)? Considering the content of the supplementary material, I get the impression that cladding/exterior wall cover was neglected which would be a very rough simplification both for LCA and LCC. If voluminous bio-based materials are used this requires also more material for the substructure. Was this taken into account? An insulation thickness of 70 cm is likely to be impossible due to space constraints (e.g. in urban context). Wherever it IS possible it is likely to call for significant changes to the building, e.g. extension of the roof as a consequence of the thicker walls, larger windows to ensure the same amount of natural light in the interior etc. Were these aspects taken into account, including both the significant LCA and LCC implications of? Moreover, for a significant share of the building stock, the exterior appearance must not be changed (heritage protection). Was this considered?

Thank you for the comments! To make sure it becomes clear that substructures, cladding and exterior wall covers were included, we have added another section in the Supplementary information where the information on the constructive details is presented. The material for substructure as solid wood was included in this study wherever necessary. Thick insulation indeed might be restricted due to space constraints however, in several Swiss cantons, the cost of space occupied by thick insulation is subsidized.

Editorial Note: Photos in Figure 1 below reproduced with permission from Atelier Werner Schmidt

Considering the thicker walls and consequent amount of natural light, larger windows installation is not taken into account in this work. Nevertheless, it is worth noting that there are several construction methods available that do not require an increase in window size. For instance, the building below that was constructed in Switzerland. In this case, slanted wooden racks are installed to achieve the necessary amount of daylight.

Figure 1: Strawbale renovation in Graubunden, architectural buro Atelier Werner Schmidt

Within our study, our focus is solely on buildings that do not fall under the category of heritage preservation. This approach stems from the fact that approximately 14% of buildings within the EU-27 were constructed before 1919 and about 12% between 1919 and 1945. Consequently, our analysis encompasses the remaining 74% of the total building stock (Ascione et al., 2015).

4. In the case of conventional materials, the insulation thickness of the optimal solutions is amazingly low while the thickness of the bio-based insulation materials is surprisingly high (very striking also acc. to Fig.6 of the appendix). Is this the outcome of the LCC or the LCC calculations? Is the reason for this outcome explained anywhere? (I don't think so) Could it be that the cost of the bio-based solutions is significantly underestimated (next to ignoring other constraints) which would explain the very large insulation thicknesses and the high energy savings?

Thank you for the questions. The costs for the biobased materials as well as labor were collected directly from the manufacturers. The optimization outcome that can be seen in Figure 6 in the Supplementary information is the pareto optimal for both LCCA and LCA where the same weight is given to both objectives. One of the possible reasons of low insulation thickness of conventional materials and high thickness of bio-based materials is the embodied carbon. The rationale behind the optimal insulation thickness can also be attributed to the resulting carbon intensity of the electricity grid. Switzerland's electricity grid, for the most part, maintains a commendably low carbon footprint (0.1 kgCO₂eq./kWh). This, in turn, leads to a preference for thinner insulation when using carbon-intensive materials. However, bio-based materials possess significantly reduced embodied carbon and the consequent carbon payback time is much shorter when comparing to the payback of the conventional materials. Therefore, think insulations still pay-off.

This is now also added to the discussion section of the paper.

5. Why is window replacement not recommended by the model while other models do so and while window replacement is also quite common in reality?

Thank you for the question. The reason for not recommending window replacement can be attributed to two primary factors: high embodied emissions and high investment costs. The manufacturing process of windows typically requires energy-intensive operations and the use of materials with significant carbon footprints such as glass and PVC/aluminum frames.

6. Which energy prices were assumed? Do the operational cost for heating in the supplementary material refer to 1 kWh of final energy or 1 kWh of useful energy (heating service)? Do the energy prices for fossil fuels include the CO₂ levy? In any case, the range of energy prices seems very/too narrow.

Thank you for the question. The efficiency of the systems such as boilers and heat pumps was included in the analysis and can be seen in Table 2 of the Supplementary information. The prices refer to the final energy (the energy delivered to the building). The price range for heating is based on a desktop study and the sources are listed in Table 2 of the Supplementary information. The energy prices of fossil fuels do not include CO₂ levy.

7. Which end-of life waste management treatment technology was assumed for conventional and non-conventional materials? How does this life cycle stage contribute to the final results?

Thank you for the comments. The end of life values were included based on the Swiss database for embodied impacts of construction materials and EPDs (Stolz & Frischknecht, 2016). A distinction between end of life and production stages was not done in the calculations – only the sum was used.

8. Since existing buildings are mostly heated with oil and gas, the yellow areas (representing non-renovated buildings) would be expected to be much wider in Fig.4.

Thank you for the comment. The uncertainty of the outcomes, in this case Q_h and LCCA is represented by the combination of uncertainties of the input parameters. The range and distribution of the inputs are based on the literature review and available data. The parameters' description is shown in Table 2 of the Supplementary information.

9. Which emission intensity was assumed for the electricity used for operating heat pumps? In more detail, do the values in the supplementary material refer to 1 kWh of final energy or 1 kWh of useful energy? Is only domestically produced electricity and wood assumed to be used? (in winter, a significant share of the electricity is imported).

Thank you for the question. Similar to the prices discussed above, the values refer to final energy. The conversion from useful energy (the heating demand Q_h) to final energy is done by using global factors for the efficiency of the heating systems. To assess the impact of the CO₂ intensity and cost of the electricity, several future scenarios were developed based on the strategy of Switzerland to phase out nuclear plants (Swiss Federal Office of Energy, 2012). The supplementary information now includes a dedicated section providing a comprehensive and detailed description of these scenarios.

10. The data sheet in the supplementary material lacks clarity, e.g. is "Cost gas boiler" = investment cost?. If so, boiler sizing does not seem to depend on the insulation level which would be questionable. Similarly, mustn't the embodied impact of the boilers (in kg CO₂ eq.) depend on their size (and in turn depend on the insulation level)?

Thank you for the comment. The range pertaining to gas boilers, wood boilers, and heat pumps is established based on the standard size commonly found in multi-family houses in Switzerland. Inclusion of costs involves an additional +/-20%, as recommended by the local standard (SIA 480, 2016). During the initial sensitivity analysis, neither the parameters of embodied emissions nor the costs of the heating system were identified as influential. The wood and gas boilers provide high-temperature heating, meaning the originally installed heat distribution system in the building usually do not have to be replaced. In contrast, in the case of heat pumps, a shift towards low-temperature heating usually mandates the replacement of the distribution system, which in turn might contribute to final costs and GHG. This was accounted for in the model and the amount and size of radiators as well as consequent cost and embodied carbon depend on the system size when low temperature heating is applied. However, as it can be seen from the result, this did not have a significant impact towards overall costs and environmental impact.

11. The discussion section refers to a follow-up study on optimal renovation scenarios but isn't this the content of the present paper?

Thank you for the comments, we agree, the explanation might not be clear. Previous work identified the methodology for robust optimal solutions for individual buildings considering separately conventional and non-conventional materials. In this study, we apply the methodologies to representative buildings from various construction periods in Switzerland that currently require building renovation. The aim is to determine whether the outcomes of the previous work can be scaled up to the whole building stock.

12. Were always individual heat pumps assumed (this is not always possible due to space and noise constraints and for aesthetic reasons)?

Thank you for the relevant question. Yes, only individual residential heat pumps were considered in this work. This is now added as a limitation point of the paper in the discussion section.

13. In many large urban agglomerations, heating with wood is restricted or forbidden. It seems this was not considered.

Thank you for the valuable comment. Indeed, in densely populated areas, the concentration of emissions from multiple wood boilers can significantly impact air quality and human health. The carbon intensity value of individual boilers fueled by gas or wood pellets is similar to that of district heating, therefore, the results of the individual boiler can be projected to the district heating, which does not produce pollutants harmful for the human health. It has also been demonstrated that the volume of particulate emissions is strongly contingent on the quality of the combustion process. The modern automatic boilers of the European variety exhibit substantially reduced pollutant levels, and are supported by European countries (Křůmal et al., 2023). This has now been added to the discussion section in the main content of the paper.

14. Does the analysis focus on space heating or is hot water supply also included?

In this study, only the heating demand was considered. The rationale behind this approach was to analyze the impact of a renovation scenario on heating demand while it does not the hot water consumption.

15. How exactly is the functional unit defined? (the functional unit should describe in quantitative terms the primary function/s fulfilled)

Thank you for the question. The functional unit refers to the use of the building over a reference study period (RSP) of 60 years. Metrics – kgCO₂eq./m², year. This is now also added to the main content of a paper.

16. If the large-scale use of bio-based materials is seriously recommended both as insulation material and for heating, the availability of these types of materials should be compared to the potential demand

Thank you for the comment. Regarding to the use of bio-based materials as insulation, the potential availability for meeting the demand of the building stock has been demonstrated to be substantial in Europe (Göswein et al., 2021). If straw is utilized as an insulation material, it has the capability to offset up to 3% of the overall greenhouse gas (GHG) emissions from all sectors by sequestering carbon within the building envelope (Pittau et al., 2019).

With regards to the availability of wood as an energy source for wood boilers or district heating systems, a recent report provides estimates for wood energy potential in Switzerland (Thees et al., 2017). The range varies from 2,500 to 25,100 GWh/year, depending on the intensity of wood extraction, ranging from sustainable to maximum theoretical levels. The theoretical potential represents the total amount of wood available as an energy source, while the sustainable potential takes into account ecological, economic, legal, and political constraints. When considering an optimal solution that utilizes wood as an energy carrier alongside thick bio-based insulation, an estimated 24,000 GWh/year would be required

to scale up to the entire building stock. When taking into account economic, ecological, and social restrictions, only 11% of the total demand could be covered. This implies that a combination of wood boilers (and/or district heating with wood as an energy carrier) and heat pumps would be necessary to upscale the solution proposed by this study to the entire building stock.

The discussion on the availability of the bio-based materials have now been added to the paper.

17. Given surging prices for energy, materials and equipment as well high inflation rates in Europe, at least some words of warning would be required. This also concerns the discount rate which would need to be higher for high inflation rates.

Thank you for the valuable comment. The discussion on future energy prices and discount rate have been added to the discussion section of the paper.

References:

- Ascione, F., Cheche, N., De Masi, R. F., Minichiello, F., & Vanoli, G. P. (2015). Design the refurbishment of historic buildings with the cost-optimal methodology: The case study of a XV century Italian building. *Energy and Buildings*, *99*, 162–176.
<https://doi.org/10.1016/j.enbuild.2015.04.027>
- Atanasiu, B., Despret, C., Economidou, M., Maio, J., Nolte, I., & Rapf, O. (2011). Europe's buildings under the microscope. In *Buildings Performance Institute Europe (BPIE)*. <https://doi.org/ISBN:9789491143014>
- European environment agency. (2023). *Greenhouse gas emission intensity of electricity generation in Europe*. <https://www.eea.europa.eu/ims/greenhouse-gas-emission-intensity-of-1>
- Göswein, V., Reichmann, J., Habert, G., & Pittau, F. (2021). Land availability in Europe for a radical shift toward bio-based construction. *Sustainable Cities and Society*, *70*(April), 102929.
<https://doi.org/10.1016/j.scs.2021.102929>
- Křůmal, K., Mikuška, P., Horák, J., Jaroch, M., Hopan, F., & Kuboňová, L. (2023). Gaseous and particulate emissions from the combustion of hard and soft wood for household heating: Influence of boiler type and heat output. *Atmospheric Pollution Research*, *14*(7).
<https://doi.org/10.1016/j.apr.2023.101801>
- Pittau, F., Lumia, G., Heeren, N., Iannaccone, G., & Habert, G. (2019). Retrofit as a carbon sink: The carbon storage potentials of the EU housing stock. *Journal of Cleaner Production*, *214*, 365–376.
<https://doi.org/10.1016/j.jclepro.2018.12.304>
- SIA 480. (2016). *Wirtschaftlichkeitsrechnung für Investitionen im Hochbau*.
- Stolz, P., & Frischknecht, R. (2016). *Koordinationskonferenz der Bau- und Liegenschaftsorgane der öffentlichen Bauherren (KBOB)*.
<https://www.kbob.admin.ch/kbob/de/home/publikationen/nachhaltiges-bauen.html>
- Streicher, K. N., Padey, P., Parra, D., Bürer, M. C., Schneider, S., & Patel, M. K. (2019). Analysis of space heating demand in the Swiss residential building stock: Element-based bottom-up model of archetype buildings. *Energy and Buildings*, *184*, 300–322.
<https://doi.org/10.1016/j.enbuild.2018.12.011>
- Swiss Federal Office of Energy. (2012). *Die Energieperspektiven für die Schweiz bis 2050 - Energienachfrage und Elektrizitätsangebot in der Schweiz 2000 - 2050 - Ergebnisse der modellrechnungen für das Energiesystem*. 1–842.
http://www.bfe.admin.ch/themen/00526/00527/06431/index.html?lang=de&dossier_id=06421
- Thees, O., Burg, V., Erni, M., Bowman, G., & Lemm, R. (2017). Biomassepotenziale der Schweiz für die energetische Nutzung, Ergebnisse des Schweizerischen Energiekompetenzzentrums SCCER BIOSWEET. *Eidg. Forschungsanstalt Für Wald, Schnee Und Landschaft WSL, WSL Berichte, Heft 57*, 1–299.

<https://www.dora.lib4ri.ch/wsl/islandora/object/wsl%3A13277/datastream/PDF/view>

REVIEWER COMMENTS

Reviewer #1 (Remarks to the Author):

I see the authors have taken substantial revision of the manuscript. Following are some still remaining issues need to be addressed.

1. I still wonder the representative of your case studies. How the selected 6 buildings can represent the existing Swiss buildings? I found they are from different construction period. What is the characteristic of buildings in the 6 different period? Why the N.5 building show distinct trend in Fig. 2 and Fig.3 and what findings can you draw from this part.

2. Also about the representativeness of renovation technologies, how are the technologies selected? What do you mean by saying "6 buildings represent 70% of the existing building stock"? How do you get the 70%?

3. The quality of figures can not meet NC standard. (1) I would not suggest to show Fig.1 ahead of corresponding content; (2) Where are the conventional and non-conventional materials displayed in Fig.3? (3) Quality of other figures and tables need to be improved.

4. I would not agree that uncertainty analysis is the key scientific contribution. Can your research give methodology reference for renovation strategy selection to European and other countries?

Reviewer #2 (Remarks to the Author):

The authors have answered very well to my comments and they have made a major effort to improve the manuscript. I therefore agree with the publication of this paper.

Having said so, it would be good if the authors nevertheless mentioned that the assessment is based on LCI data which include both production and end-of-life without allowing to distinguish the two phases (this can have some impacts on the findings). In addition, it is recommended to define the functional precisely, in line with good practice in LCA.

Response to referees

Article title - Strategies for robust renovation of residential buildings – A Swiss focus

REVIEWER COMMENTS

Reviewer #1 (Remarks to the Author):

I see the authors have taken substantial revision of the manuscript. Following are some still remaining issues need to be addressed.

1. I still wonder the representative of your case studies. How the selected 6 buildings can represent the existing Swiss buildings? I found they are from different construction period. What is the characteristic of buildings in the 6 different period? Why the N.5 building show distinct trend in Fig. 2 and Fig.3 and what findings can you draw from this part.

Thank you for your valuable comment regarding the representativeness of the selected case studies. We understand the need for clarity in this aspect.

The selected buildings, accounting for approximately 70% of the residential buildings constructed before 1990, were identified through the eRen project. These case studies were derived from 15 building models, representing the multi-family house residential building stock in Western Switzerland. This selection is based on data from 193 buildings in Fribourg, Vaud, and Geneva cantons, collectively encapsulates between 72% to 89% of Switzerland's dwelling count according to the Federal Statistical Office (Federal Statistical office, 2023).

Prior research extensively characterized the thermal performance and retrofit status of Swiss residential buildings using Swiss Cantonal Building Energy Certificates and highlighted the variability within the building stock (Streicher et al., 2018). The case studies presented in our work notably represent buildings with the lowest energy performance, predominantly lacking insulation, and cover most of operational emissions of buildings in Switzerland while being supplied by oil and gas.

To evaluate their representativeness for the entire building stock, a supplementary study was performed. This involved scaling up the energy consumption of our selected building representatives in alignment with the energy reference area corresponding to each construction period (Streicher et al., 2019). Comparing these estimates against actual heating energy consumption data for buildings constructed before 2000 (Swiss Federal Office of Energy, 2012) revealed a minor deviation of merely 4%. This analysis allows the generalizability of our study results to the building stock in Switzerland.

Indeed, notable results can be observed for building 5 regarding the life cycle cost analysis. This can be linked to a higher variability of the heating demand of a non-renovated building and consequent costs as the uncertainty was set as a percentage according to the local standard SIA 480:2016 (SIA 480, 2016).

The methodology section of the paper was adapted to clarify the representativeness of the case studies.

2. Also about the representativeness of renovation technologies, how are the technologies selected? What do you mean by saying “6 buildings represent 70% of the existing building stock”? How do you get the 70%?

Thank you for your valuable comment. To address the renovation methods and thermal performance selection, we systematically progressed from uninsulated states to achieving U-values target of SIA

380/1 (0.25 W/(m²·K)), followed by reaching the punctual requirement for subsidies from the Gebaudeprogram (0.2 W/(m²·K)), as well as targeting ≤ 0.17 W/(m²·K) in an extreme scenario according to Minergie certification. Additionally, we have added the cases falling between these specified requirements to precisely address the optimal renovation. The non-renovated case for all the components was also considered. The insulation thickness was adapted in accordance with market availability. The insulation thickness varies for each case study due to the varying heat loss coefficient of the initial elements.

Regarding material selection, our study comprehensively considers a diverse range of both conventional and non-conventional materials. Conventional options encompass widely-used materials like EPS, XPS, cellular insulation, wood fibers, rockwool, and glasswool. Alongside these, we have evaluated non-conventional materials such as straw bale, hemp mat, and hempcrete, aiming to present a comprehensive overview of available options considering different thermal properties as mentioned in the recent paper (Casini, 2020).

The description of the approach was added to the Supplementary information of the paper.

We believe that the topic of building representativeness is addressed in our response to the first question. If you need further clarification or additional details, please, let us know.

3.The quality of figures can not meet NC standard. (1) I would not suggest to show Fig.1 ahead of corresponding content; (2)Where are the conventional and non-conventional materials displayed in Fig.3? (3)Quality of other figures and tables need to be improved.

Thank you for the comment, we substantially reworked all the figures and tables of the paper.

4.I would not agree that uncertainty analysis is the key scientific contribution. Can your research give methodology reference for renovation strategy selection to European and other countries?

Thank you for your comment. Indeed, the scientific contribution of the paper is not only the uncertainty analysis but also a methodology development that includes integrated LCCA and LCA, uncertainty quantification and sensitivity analysis. The methodology can be applied to any context where the heating demand is prevailing in the energy consumption of a building.

We have reframed the scientific contribution of the paper as following:

“The scientific contribution of the paper is the development of a methodology to identify the optimal and robust building renovation scenario including the associated uncertainty for the building representatives of all the construction periods in Switzerland that currently require renovation. The methodology includes both conventional and bio-based materials and can be applied to the countries where the prevailing energy consumption is heating. “

Reviewer #2 (Remarks to the Author):

The authors have answered very well to my comments and they have made a major effort to improve the manuscript. I therefore agree with the publication of this paper.

Having said so, it would be good if the authors nevertheless mentioned that the assessment is based on LCI data which include both production and end-of-life without allowing to distinguish the two phases (this can have some impacts on the findings). In addition, it is recommended to define the functional precisely, in line with good practice in LCA.

Thank you very much for your positive feedback. The discussion on the combined production and end of life stages is now included in the limitations of the paper. The functional unit is also described in more details in the main content of the paper.

The following was added in the limitation chapter:

“Another limitation of the study is that the current assessment is based on LCI data encompassing both production and end of life stages of LCA. Nevertheless, the outcomes do not to differentiate the distinct impacts of these stages when considered individually, potentially influencing the study's findings.”

The functional unit was defined as follows:

“The functional unit for this study encompasses the complete life cycle of a renovation of a residential building operating over a period of 60 years, including construction, occupancy, maintenance, and potential replacement of building components. It evaluates the environmental impacts associated with the building's production, replacement, operational energy demand for heating, and end-of-life stages, considering the use of both conventional and non-conventional materials.”

Casini, M. (2020). Insulation Materials for the Building Sector: A Review and Comparative Analysis. In *Encyclopedia of Renewable and Sustainable Materials: Volume 1-5* (Vols. 1–5). Elsevier Ltd. <https://doi.org/10.1016/B978-0-12-803581-8.10682-4>

Federal Statistical office. (2023). *Buildings and dwellings from 2010 onwards*. <https://www.bfs.admin.ch/bfs/en/home/services/geostat/swiss-federal-statistics-geodata/population-buildings-dwellings-persons/buildings-dwellings-from-2010.html>

SIA 480. (2016). *Wirtschaftlichkeitsrechnung für Investitionen im Hochbau*.

Streicher, K. N., Padey, P., Parra, D., Bürer, M. C., & Patel, M. K. (2018). Assessment of the current thermal performance level of the Swiss residential building stock: Statistical analysis of energy performance certificates. *Energy and Buildings*, 178, 360–378. <https://doi.org/10.1016/j.enbuild.2018.08.032>

Streicher, K. N., Padey, P., Parra, D., Bürer, M. C., Schneider, S., & Patel, M. K. (2019). Analysis of space heating demand in the Swiss residential building stock: Element-based bottom-up model of archetype buildings. *Energy and Buildings*, 184, 300–322. <https://doi.org/10.1016/j.enbuild.2018.12.011>

Swiss Federal Office of Energy. (2012). *Die Energieperspektiven für die Schweiz bis 2050 - Energienachfrage und Elektrizitätsangebot in der Schweiz 2000 - 2050 - Ergebnisse der modellrechnungen für das Energiesystem*. 1–842. http://www.bfe.admin.ch/themen/00526/00527/06431/index.html?lang=de&dossier_id=06421

REVIEWERS' COMMENTS

Reviewer #1 (Remarks to the Author):

I think the authors have well addressed my previous comments. While the Figure quality still have potential to meet higher standard. Try to use more outstanding color and unify colors among Figures.

Response to referees

Article title - Strategies for robust renovation of residential buildings in Switzerland

REVIEWER COMMENTS

Reviewer #1 (Remarks to the Author):

I think the authors have well addressed my previous comments. While the Figure quality still have potential to meet higher standard. Try to use more outstanding color and unify colors among Figures.

Thank you for your valuable comment. Regarding the Figure quality and color scheme, we tried to maintain consistency in the use of colors across all figures to reflect the specific materials they belong to. Notably, Figs 3–6 are consistently depicted in orange as they relate to a wood boiler. Our aim was to establish a color scheme that provides equal visibility to all material categories while ensuring accessibility for color-blind users.

In response to your suggestion, we have adjusted the color scheme to employ darker tones, aiming to enhance overall figure visibility while maintaining the equal representation of different material types. We trust that these adjustments address your concerns, and we remain open to any further suggestions you may have.